# Gene Expression-Based Cancer Classification for Handling the Class Imbalance Problem and Curse of Dimensionality

**DOI:** 10.3390/ijms25042102

**Published:** 2024-02-09

**Authors:** Sadam Al-Azani, Omer S. Alkhnbashi, Emad Ramadan, Motaz Alfarraj

**Affiliations:** 1SDAIA-KFUPM Joint Research Center for Artificial Intelligence, King Fahd University of Petroleum and Minerals (KFUPM), Dhahran 31261, Saudi Arabia; motaz@kfupm.edu.sa; 2Information and Computer Science Department, King Fahd University of Petroleum and Minerals (KFUPM), Dhahran 31261, Saudi Arabia; omer.khnbashi@kfupm.edu.sa (O.S.A.); eramadan@kfupm.edu.sa (E.R.); 3Electrical Engineering Department, King Fahd University of Petroleum and Minerals (KFUPM), Dhahran 31261, Saudi Arabia

**Keywords:** cancer detection and diagnosis, gene expression, feature selection, class imbalance

## Abstract

Cancer is a leading cause of death globally. The majority of cancer cases are only diagnosed in the late stages of cancer due to the use of conventional methods. This reduces the chance of survival for cancer patients. Therefore, early detection consequently followed by early diagnoses are important tasks in cancer research. Gene expression microarray technology has been applied to detect and diagnose most types of cancers in their early stages and has gained encouraging results. In this paper, we address the problem of classifying cancer based on gene expression for handling the class imbalance problem and the curse of dimensionality. The oversampling technique is utilized to overcome this problem by adding synthetic samples. Another common issue related to the gene expression dataset addressed in this paper is the curse of dimensionality. This problem is addressed by applying chi-square and information gain feature selection techniques. After applying these techniques individually, we proposed a method to select the most significant genes by combining those two techniques (CHiS and IG). We investigated the effect of these techniques individually and in combination. Four benchmarking biomedical datasets (Leukemia-subtypes, Leukemia-ALLAML, Colon, and CuMiDa) were used. The experimental results reveal that the oversampling techniques improve the results in most cases. Additionally, the performance of the proposed feature selection technique outperforms individual techniques in nearly all cases. In addition, this study provides an empirical study for evaluating several oversampling techniques along with ensemble-based learning. The experimental results also reveal that SVM-SMOTE, along with the random forests classifier, achieved the highest results, with a reporting accuracy of 100%. The obtained results surpass the findings in the existing literature as well.

## 1. Introduction

A gene is a specific segment of chromosome DNA containing the genetic material that guides all living things’ growth, development, and function. Genes are the fundamental units of inheritance. They are responsible for many of our genetic characteristics and traits, including eye color, body size, and risk regarding certain diseases [1].

There are many different types of genes, including structural genes that code for proteins, regulatory genes that control the expression of other genes, and noncoding genes that do not code for proteins but have other essential functions.

Gene expression is the process by which information is encoded in gene synthesis proteins, which then perform a specific function in the cell. This involves measuring the activity of thousands of genes in tissue samples, thereby providing detailed insight into the genetic changes occurring in healthy and unhealthy cells [1,2].

Clinical and pathological information may be incomplete or misleading for detecting and diagnosing cancer [1]. Cancer detection and diagnosis can be objective and highly accurate using microarrays, which have the potential to provide clinicians with the information needed to choose the most appropriate form of treatment [2]. Cancer detection and diagnosis using gene expressions as a classification problem originates from human acute leukemia classification as a test case [3].

The major challenge in microarray data is the curse of dimensionality such that it includes a large number of genes with a small number of samples. To alleviate this limitation, researchers proposed several feature selection techniques that can select the most significant genes and investigated several statistical and machine learning classification methods.

Another critical concern related to microarray datasets is the emergence of the class imbalance problem, which is characterized by a significant difference in the number of samples per class [4]. Notably, this particular issue, which pertains to the class imbalance problem, has not been previously tackled in the context of cancer classification utilizing gene expression data.

This study proposes a method for classifying cancer using gene expression datasets while addressing the issues of class imbalance and the curse of dimensionality. Three publicly available microarray datasets used for this study suffer from the problem of class imbalance and the dimensionality curse. They are binary and multiple classes. This study also presents a method for selecting the most significant genes by combining two feature selection techniques. Multilayer perceptron (MLP), support vector machines (SVMs), and random forests classifiers are applied and compared. Furthermore, a uniform k-NN classifier based on the majority voting method is proposed, in which the underlying k-NN models are built through learning different values of k. The main contributions of the paper can be summarized as follows:This paper proposes a method to classify cancer using gene expression datasets, thereby effectively tackling the challenges of class imbalance and the dimensionality curse in the context of three publicly available microarray datasets. The proposed method applies to both binary and multiple class classification problems, thereby making it versatile and suitable for a wide range of cancer classification scenarios.This study presents an empirical study for evaluating several oversampling techniques along with ensemble-based classification algorithms.This paper introduces a method that combines two feature selection techniques to identify the most significant genes, which are crucial for accurate cancer classification.This study provides a comprehensive in-depth empirical analysis of the proposed approach using different basic and ensemble classifiers, different evaluation methods, and several evaluation measures.

Adhering to the format of this journal, the rest of this paper is organized as follows. The remainder of this section provides a review of the most related works. The results and discussion are provided in Section 2. The proposed research method is presented in Section 3. Lastly, the conclusion and future directions are presented in Section 4.

### Related Works

Lee et al. [5] used a neural network-based finite impulse response extreme learning machine (FIR-ELM) to classify leukemia and colon tumors. The proposed method (FIR-ELM) performed better than other classifiers for the leukemia dataset. However, SVMs performed better for the colon dataset, which was followed by the FIR-ELM.

Lotfi and Keshavarz [6] presented an approach called PCA-BEL based on principle component analysis and a brain emotional learning network for the classification of gene expression microarray data. The pros of using BEL include its low computational complexity.

Rathore, Iftikhar, and Hussain [7] used two feature selection techniques in the sequence. The first feature selection technique, chi-square, takes the whole dataset and selects the discriminative gene subset, which is then used as input for the second feature selection technique, mRMR, which selects the most discriminative gene subset among them. They reported that the proposed technique achieved classification rates and performed better than the individual techniques.

Another approach that utilizes the nature of variations in gene expressions to classify colon gene samples into normal and malignant classes was conducted by Rathore, Hussain, and Khan [8]. They presented a majority voting-based ensemble of SVMs. Experimental results revealed that the ensemble classifier improved the results compared to the single classification techniques.

Bouazza et al. [9] reported that the signal-to-noise ratio (SNR) feature selection technique is the most trusted technique for selecting the genes from three different datasets when compared to Fisher, T-Statistics, and ReliefF. Banka and Dara [10] proposed a Hamming distance-based binary particle swarm optimization (HDBPSO) algorithm to select the most significant gene subsets.

Simjanoska, Bogdanova, and Popeska [11] analyzed gene expression to classify colon carcinogenic tissue. Illumina HumanRef-8 v3.0 Expression BeadChip microarray technology was utilized to conduct the gene expression profiling, which contained 26 colorectal tumors and 26 colorectal healthy tissues. An original methodology containing several steps was developed for biomarker detection, which included data preprocessing, statistical analysis, and modeling the a priori probability for all significant genes. It was reported that Bayes’ theorem performed better than SVMs and BDT. These findings were justified due to the realistic modeling of the a priori probability of Bayes’ theorem. However, such a method is somewhat complicated. The a priori probability model generated in [11] was then employed by Bogdanova, Simjanoska, and Popeska [12] using gene expression with the Affymetrix Human Genome U133 Plus 2.0 Array; the results revealed poor distinctive capability regarding the biomarker genes. That means that the a priori probability model is platform-dependent. This finding confirms what was concluded in Wong, Loh, and Eisenhaber [13] where they stated that each platform requires different statistical treatment [13]. Simjanoska, Madevska Bogdanova, and Popeska [14] generated a statistical approach for obtaining gene expression values obtained from Affymetrix using a similar methodology to [11]. The findings revealed that excellent results were achieved using Bayes’ theorem when an appropriate preprocessing methodology was followed. The results reported in [14] were then improved in the work of Simjanoska and Bogdanova [15]. They proposed a filtering gene method to select the most essential biomarkers. It is called the leave-one-out method and is based on iterative Bayesian classification.

Tong et al. [16] introduced a genetic algorithm (GA)-based ensemble SVM classifier constructed using gene pairs (GA-ESP). The base classification methods (SVMs) of the ensemble system were trained on various informative gene pairs. These gene pairs were selected using the top-scoring pair (TSP) criterion. GA was then employed to select an optimized combination of base classifiers. The applicability of the proposed approach was evaluated on several cancer datasets in both binary class and multiple class datasets.

Cao et al. [17] presented a novel fast feature selection method based on multiple SVDD, which is called MSVDD-RF. Insignificant features are eliminated recursively in this method. It was applied to multiple class microarray data to detect different cancers. Table 1 summarizes the reviewed studies in terms of the type of cancer, the used and proposed techniques, the feature selection techniques, and the datasets used. Modern techniques for gene expression-based classification and diagnosis are reviewed in [18,19].

Liu et al. [20] proposed a combined solution to classify DNA methylation imbalance data for cancer risk prediction using the the synthetic minority oversampling technique (SMOTE) and Tomek Links methods. They looked only at genes whose mutations were associated with cancer, which were obtained through sources such as the Catalog of Somatic Mutations in Cancer (COSMIC) and the Clinical Interpretation of Cancer Variants (CIViC). This research method was applied to TCGA DNA methylation data for 28 different cancer types, thereby demonstrating superior performance in classifying patient samples based on the combined use of SMOTE and SMOTE sampling methods. T-Link is a key innovation aspect of the research.

Pasksoy et al. [21] studied colon cancer using artificial intelligence and genomic biomarkers using biomarker candidate genes for colon cancer, and they developed a model for predicting the disease based on these genes. They used a dataset with gene expression levels from 62 samples (22 healthy and 40 tumor tissues). In order to overcome the class imbalance in the dataset, SMOTE was applied. In addition, the LASSO feature selection method was used to select the 13 most-important genes associated with colon cancer. Lastly, three classification methods, random forests (RFs), decision tree (DT), and Gaussian naive Bayes, were used to develop the prediction model.

Arafa et al. [22] addressed the challenges of high dimensionality and class imbalance in gene expression datasets, which can negatively impact the performance of classifiers used in cancer classification. They presented a model called RN-Autoencoding to classify imbalanced cancer genomic data. The autoencoder was used for feature reduction to solve the dimensionality problem and to produce newly extracted data with lower dimensionality. Then, the reduced noise synthetic minority sampling technique (RN-SMOTE) was applied to handle the class imbalance in the extracted data. RN-SMOTE uses the density-based spatial clustering of applications with noise (DBSCAN) algorithm to detect and remove noise after oversampling an imbalanced dataset via SMOTE. The results showed that the performance of the classifiers was improved with RN-Autoencoding and outperformed the performance of the original and extracted data.

The majority of these studies employed feature selection techniques separately, with the exception of the study conducted by Rathore et al. [7]. A notable distinction is that they utilized two feature selection techniques sequentially (ChiS and mRMR), whereas our approach involves the simultaneous combination of two techniques (ChiS and IG). This difference in approach is complemented by variations in the underlying methodologies.

## 2. Results and Discussion

### 2.1. Evaluation of Oversampling Techniques with Ensemble Learning

This section provides an empirical study for evaluating several oversampling algorithms, namely, SMOTE, BorderlineSMOTE, SVM-SMOTE, and KMeanSMOTE (SMOTE-KN). The SMOTE [23] involves creating artificial samples for the minority class, thereby effectively addressing the imbalance issue without relying on simple random oversampling with replacement. Creating synthetic instances through the SMOTE algorithm depends on the resemblances within the feature space of existing minority instances. The SMOTE randomly selects data points from the minority class $x_{i}\in S_{min}$ such that $S_{min}$ is the minority class. Then, it identifies the *k*-nearest neighbors of the selected data point (the value of *k* is an integer value). This is followed by creating a new data point by linearly interpolating between the selected data point and one of its *k*-nearest neighbors. This is repeated until the desired number of artificial samples are created. Figure 1 depicts an illustration of SMOTE algorithm.

BorderlineSMOTE [24] represents a modification of the original SMOTE algorithm. It involves the identification of borderline samples, which are subsequently utilized for the generation of new synthetic samples. It has two variations: SMOTE-B1 and SMOTE-B2; both of them are evaluated in this study. SVM-SMOTE [25] is another variation of the original SMOTE. It employs an SVM classifier to identify support vectors and utilizes this information to generate new samples. Another variation of SMOTE, is KMeanSMOTE [26] which employs a K-Means clustering technique prior to the application of SMOTE. Through clustering, samples are grouped together, and new samples are generated based on the density of each cluster. They are implemented using the imbalanced-learn tool [27]. Using the oversampling techniques is typically done on the training datasets, since it is not practical to create synthetic testing data and assess models using synthetic test samples [28]. Once the oversampling techniques are applied, the quantity of training samples for every class across all datasets is adjusted to match the highest counts of the majority classes in the original datasets. A recently published dataset called Curated Microarray Database (CuMiDa) [29] was used for the empirical analysis. CuMiDa is the gene expression dataset for leukemia cancer composed of 22,283 genes and 64 samples distributed among five leukemia types presented in Table 2.

Different ensemble-based learning models were generated and evaluated using bagging, random forests, stacking, voting, and boosting. Table 3 presents a description of the evaluated ensemble learning implemented using scicit-learn [30]. As for the evaluation method, a 10-fold cross-validation approach was adopted. This approach served the dual purpose of benchmarking our results and facilitating comparisons with the existing literature. Principal Component Analysis (PCA) was additionally employed to condense the input features, limiting them to only 50 genes. Table 4 presents the obtained results. The highest results were obtained using SVM-SMOTE, along with random forests and hard voting classifiers, with each reporting 100% accuracies.

For the empirical analysis of the evaluated oversampling techniques, we also evaluated their effects and compared them with the baselines. We considered the ensemble learning classifiers without oversampling as the baselines. As depicted in Figure 2, applying oversampling techniques had positive impacts in nearly all cases for all the evaluation measures except for two cases. The first case was when applying SMOTE with the stacking-based ensemble classifier, in which there was a drop in the results by around 1.5%. The other case was when applying the SMOTE with the hard voting-based ensemble classifier.

### 2.2. Evaluation of Proposed Feature Selection Method

This section presents an empirical study of the proposed feature selection technique which involves the combination of chi-square (ChiS) and information gain (IG) methods under two scenarios: one involving the application of the oversampling technique and the other without it. The evaluation is conducted using three gene expression datasets, as described in Table 5 and Table 6. The empirical analysis employs three classifiers—namely, multilayer perceptron (MLP), sequential minimal optimization support vector machines (SMO-SVM), and random forests.

For MLP classifier, an extensive exploration of various neural network architectures was conducted. This exploration involved experimenting with different configurations, including varying the number of neurons in the hidden layers and adjusting the learning rates. The goal was to systematically analyze how these architectural parameters impact the performance of the MLP classifier in our study. In the first network configuration, the count of hidden layer neurons is determined by computing the average of the input and output dimensions such that
(1)$$\mathrm{number}\;\mathrm{of}\;\mathrm{neurons}=\frac{\mathrm{number}\;\mathrm{of}\;\mathrm{input}\;\mathrm{attributes}}{\mathrm{number}\;\mathrm{of}\;\mathrm{classes}}$$ As an illustration, if there are 300 genes in the input and seven classes in the output (representing the number of distinct classes), the number of neurons in the hidden layer would be calculated as 154. Regarding the learning rate, a fixed value of 0.3 was employed.

We explored other architectures of the MLP by varying the number of neurons in the hidden layer. Specifically, we investigated configurations with 20, 50, and 80 neurons in the hidden layer. Additionally, we examined various learning rates, including 0.1, 0.3, and 0.5, thereby considering all the possible combinations of these parameters. Every MLP configuration was thoroughly examined using a momentum value of 0.2 in conjunction with a backpropagation learning algorithm.

In the case of the SVM classifier, we employed the SMO-SVM as an optimization algorithm used for training support vector machines [31]. The SMO-SVM is a specific algorithm designed to efficiently solve the optimization problem associated with SVM training. The main idea behind SMO is to break down the large quadratic programming problem into a series of smaller subproblems that can be solved analytically and efficiently. This approach is particularly useful when dealing with large datasets and high-dimensional feature spaces. This investigation focused on assessing the performance of two distinct kernel functions: PUK and Poly-kernel. The complexity parameter (C) was set to one for both experiments. To address the multiclass problem, we applied pairwise classification, which is a technique commonly referred to as “one-vs-one”.

As ensemble learning classifiers, we applied random forests, which aggregate decision tree predictors. Each individual tree depends on the values of a random vector sampled independently, and all the trees within the forest share the same distribution [32].

Our approach involves suggesting a homogeneous ensemble of k-NN classifiers for the classification of cancers. This ensemble leverages the concept of majority voting among the predicted labels generated by individual k-NN models. The majority voting algorithm aggregates the predictions from each of the 1-NN, 3-NN, and 5-NN models separately (where 1-NN stands for one nearest neighbor, 3-NN for three nearest neighbors, and so on). Subsequently, it assigns a label to a sample by considering the most frequently occurring prediction among these models. This approach was implemented using WEKA-3.6.13 [33].

The outcomes of our experiments are consolidated in Table 7, Table 8 and Table 9 corresponding to MLP, SMO-SVM, and random forests, respectively.

Each table is structured to present the outcomes of individual feature selection without oversampling, individual feature selection with oversampling, and the proposed combined feature selection technique before and after the implementation of the oversampling technique. Additionally, each table is divided into three sections, with each section depicting the results for a specific dataset.

The most favorable performances are denoted by bold numbers, wherein we define the best performance as the one associated with the highest precision, recall, F measure, and accuracy. Subsequently, the number of selected genes was considered. In situations where the results achieved through various feature selection methods were equal, our preference leaned towards the technique with the smallest number of genes. For instance, in the Leukemia-subtype scenario, the results obtained using ChiS were identical to those achieved using the combined technique (ChiSIG). Nevertheless, we considered the outcomes obtained through the combined technique to be superior, because it excelled in terms of the number of selected genes. Specifically, while the ChiS resulted in 300 genes, the combined technique yielded a reduced set of 233 genes. This principle was applied consistently across the other cases as well.

In the case of the MLP, the most optimal performance was achieved by employing the first structure, as previously described, for all datasets. Conversely, for the SMO-SVM, the peak performance was observed when utilizing the poly kernel function for the Leukemia-subtype and Leukemia-ALLAML datasets, and the PUK kernel function for the Colon dataset.

In the Leukemia-subtype dataset, the most superior performance was achieved when utilizing our proposed feature selection method in conjunction with the SMO-SVM. Similarly, for the Leukemia-ALLAML dataset, our feature selection technique yielded the best performance when employed alongside the MLP. In the scenario of the Colon dataset, both the MLP and SMO-SVM exhibited their highest performance levels when our proposed feature selection technique was employed.

In order to assess the impact of the SMOTE techniques, we categorized its effects into three distinct types: positive influence, negative influence, and negligible influence. The experimental findings presented in Table 7, Table 8 and Table 9 illustrate that in 19 out of the 27 scenarios, the application of the SMOTE technique had a favorable impact, thereby resulting in improved outcomes. Furthermore, in three instances, the SMOTE technique yielded adverse effects, while it showed no discernible impact in five other cases.

We conducted an investigation into the sensitivity of the classifiers for the evaluated oversampling techniques. The experimental findings reveal that the random forests classifier exhibited the highest degree of positive improvements. This sensitivity analysis is visually represented in Figure 2, which demonstrates that the overall average improvement in all evaluation measures achieved through the application of the evaluated oversampling techniques was more pronounced in the case of the random forests.

We conducted additional experiments involving random forests and an ensemble of k-NN classifiers. For these experiments, we employed the ChiS feature selection method within a 10-fold cross-validation framework while applying the SMOTE. We conducted an examination to identify the most informative genes within the three datasets. Figure 3 presents the correlation between the number of these informative genes and the associated accuracies when employing the random forests classifier. In the case of the Leukemia-subtype dataset, it is evident that the random forests classifier achieved the highest accuracy, reaching 96.56%, with a gene set comprising 90 genes. Shifting to the Leukemia-ALLAML dataset, it is apparent that the random forests classifier achieved a perfect accuracy rate of 100% using a gene set containing 50 genes. Finally, in the context of the Colon dataset, it is clear that the highest accuracy achieved by the random forests classifier stood at 92.50%, and this performance was maintained with a gene set consisting of 60 genes or more. Table 10 provides an overview of the top-performing results achieved using the random forests classifier for the most significant gene sets.

The same process described above was also implemented when using an ensemble of k-NN classifiers in lieu of random forests. Figure 4 displays the relationship between the number of the most-informative genes and their corresponding accuracies when employing the ensemble of k-NN classifiers. In the case of the Leukemia-subtype dataset, the highest accuracy reached was 93.85%, which was achieved with a gene set containing 100 genes. Shifting to the Leukemia-ALLAML dataset, the highest accuracy obtained was 98.94% utilizing a gene set comprising 60 genes. Finally, in the context of the Colon dataset, it is apparent that the highest accuracy achieved stood at 92.50%, and this performance was maintained with a gene set consisting of 30 genes or more. Table 11 presents the top-performing results obtained with the ensemble of k-NN classifiers for the most-prominent gene sets.

It is imperative to contextualize our results within the broader research landscape by conducting a thorough comparison of our obtained results with those of related studies. This comparative analysis was guided by a set of well-defined criteria, as elaborated in Table 12. Importantly, our findings shined brightly in this comparative analysis, as they consistently outperformed the results reported in the most closely related works. This not only underscores the robustness of our approach but also highlights its potential to make a significant contribution to the domain.

## 3. Materials and Methods

Figure 5 depicts the high architecture of the proposed framework. It is composed of several tasks: gene expression microarray datasets preparation, addressing the imbalance issue, selecting the most significant genes, classification, and evaluation. The following subsections describe in detail all of those tasks.

### 3.1. Gene Expression Microarray Datasets Collection and Preparation

In this study, we used different benchmarking gene expression microarray datasets: Curated Microarray Database (Leukemia CuMiDa) [29], Lymphoblastic Leukemia (Leukemia-subtype) dataset [36], Leukemia-ALLAML [3], and Colon tumor [37]. Both Leukemia-ALLAML and Colon datasets are binary class datasets such that the Leukemia-ALLAML dataset includes two main types, ALL and ALM, while the Colon dataset includes normal and malignant samples. On the other hand, Leukemia CuMiDa and Leukemia-subtype are multiclass datasets covering five and seven types, respectively. The minority class for the Leukemia-ALLAML dataset is AML, with only 11 samples, while the majority class is ALL with 27 samples. The numbers of genes of the Leukemia-ALLAML dataset is 7129. Similarly, the majority class for the Colon dataset is tumor with 28 samples, while the minority class is normal with 15 samples. The number of genes in the Colon dataset is 2000. Leukemia-subtype dataset is composed of 12,558 genes in which the minority class is BCR-ABL, with only nine samples, while AML (TEL-AML1) and others represent the majority class, with 52 samples for each. Leukemia CuMiDa dataset is composed of 22,283 genes; the minority class is Bone_Marrow_CD34 with 8 samples while the majority class is AML with 26 samples. More details for the process of gene expression data extraction can be found in the original papers on the Leukemia CuMiDa dataset [29], Leukemia-subtype dataset [36], Leukemia-ALLAML [3], and Colon tumor [37].

Table 2 and Table 5 provide statistical summaries of the multi-class datasets (Leukemia CuMiDa and Leukemia-subtype ), while Table 6 offers a statistical overview of the binary class datasets (Leukemia-ALLAML dataset and Colon tumor dataset).

The genes underwent a normalization process employing the min–max normalization method, which scales their values to a standardized range between zero and one. This normalization technique ensures that gene expression values are uniformly represented within the specified range, thereby facilitating fair comparisons and reducing the impact of variations in gene expression scales across different datasets or experiments.
(2)$$\tilde{x}=\frac{x-x_{\min}}{x_{\max}-x_{\min}}$$

### 3.2. Addressing the Imbalance Issue

Imbalanced datasets can pose challenges for machine learning algorithms. Handling such datasets requires specialized techniques to ensure that the model does not become biased towards the majority class. Different approaches have been proposed to deal with the imbalance data issue in the literature including data level, algorithm level, and a hybrid of data and algorithm level [19]. Data level-based techniques (resampling techniques) are classified as oversampling, undersampling, or a hybrid of oversampling and undersampling techniques [38]. On the other hand, the algorithm-based techniques for imbalanced datasets address the imbalance issue by allowing for the assignment of different class weights to penalize misclassification of the minority class more heavily [39]. Ensemble learning methods represent another category of algorithmic approaches used to address the issue of class imbalance. They can improve the model’s ability to handle imbalanced data by combining multiple models [40]. The issue of high imbalance in this study is mitigated by creating synthetic samples for the minority class at the data level. SMOTE [23] stands for synthetic minority oversampling technique, which is an algorithm designed to tackle the challenge of imbalanced datasets.

Table 13 shows the majority and minority class distributions of the datasets.

### 3.3. Feature Selection

Feature selection is a crucial step in the data preprocessing pipeline, thereby helping to improve model performance, reduce complexity, and enhance interpretability by selecting the most informative and relevant features for a given machine learning task [41]. These methods can be classified as filter methods, which rank features based on statistical measures (e.g., correlation, chi-squared, mutual information) and select the top-ranked features without involving machine learning algorithms. Another type of feature selection method is the wrapper method, which selects subsets of features and evaluates them using machine learning algorithms such as recursive feature elimination [42,43].

We explored two feature selection approaches, chi-square (ChiS) and information gain (IG), which assess the importance of attributes by assigning them to ranks. The chi-square method assesses attributes in relation to the class by calculating the chi-square statistic, while the information gain (IG) feature selection technique measures the value of an attribute by quantifying the information gained concerning the class. Our approach involves combining both chi-square and information gain methods to select the most meaningful genes.

Both feature selection techniques, ChiS and IG, assign a ranking to each gene within the datasets under evaluation. We simply established a threshold of zero for both methods, which resulted in the exclusion of genes with negative rankings. In this study, we introduced a feature selection technique called ChiSIG, which combines the ChiS and IG methods. ChiSIG identifies genes that are mutually selected by both of these techniques. As a consequence, a reduced set of selected genes is obtained, thereby assuming that these genes hold more significance than those chosen by the individual techniques.

Table 14 displays the count of both complete and selected genes in the datasets following the application of feature selection techniques.

### 3.4. Classification Phase

In our study, we employed three distinct classifiers, namely, multilayer perceptron (MLP), sequential minimal optimization support vector machines (SMO-SVM), and random forests, for the purpose of classifying different types of cancer. We conducted a comprehensive comparison among these classifiers to assess their performance in cancer type detection from gene expression data. We also introduced a homogeneous ensemble of k-NN classifiers that relies on majority voting.

### 3.5. Evaluation Metrics

In this study, we utilized widely recognized performance metrics, which include precision, recall, F measure, and accuracy. These metrics involve the computation of specific quantities: true positives (*TP*) representing correctly classified positive samples, false negatives (*FN*) representing positive samples that are misclassified, true negatives (*TN*) representing correctly classified negative samples, and false positives (*FP*) representing negative samples that are misclassified.
(3)$$Precision=\frac{TP}{TP+FP}\times 100$$
(4)$$Recall=\frac{TP}{TP+FN}\times 100$$
(5)$$F1=\frac{\left(2\times Precision\times Recall\right)}{\left(Precision+Recall\right)}\times 100$$
(6)$$Accuracy=\frac{TP+TN}{TP+FP+TN+FN}\times 100$$

## 4. Conclusions

This study introduced an approach for the classification of various cancer types utilizing gene expression datasets. It addresses two primary challenges: class imbalance and the curse of dimensionality commonly found in gene expression microarray datasets. The challenge of class imbalance was tackled in this study at two levels: the data level and the algorithm level. At the data level, oversampling techniques were employed, while at the algorithm level, ensemble learning was utilized. Furthermore, we tackled the curse of dimensionality problem by proposing a combined feature selection technique of two well-established feature selection techniques, namely, chi-square and information gain.

An empirical evaluation was conducted on four publicly available biomedical datasets (Leukemia-subtype, Leukemia-ALLAML, Colon tumor, and Leukemaia CuMiDa) afflicted with the class imbalance and the curse of dimensionality. The experimental findings highlight that the suggested feature selection method consistently attained the highest performance across nearly all scenarios. Furthermore, the application of the oversampling techniques enhanced the outcomes in nearly all cases. In addition, the SVM-SMOTE oversampling technique, along with random forests, achieved the highest results compared to the other evaluated oversampling and ensemble learning techniques.

Additionally, this research includes an examination of how sensitive the adopted classifiers are to the SMOTE technique. The experimental outcomes reveal that the SMO-SVM and random forests methods exhibited a higher positive sensitivity to the impact of the SMOTE compared to the MLP.

In future work, we will explore the use of RNA-seq data to align with the latest trends in genomics research using advanced deep learning techniques.

## Figures and Tables

**Figure 1 ijms-25-02102-f001:**
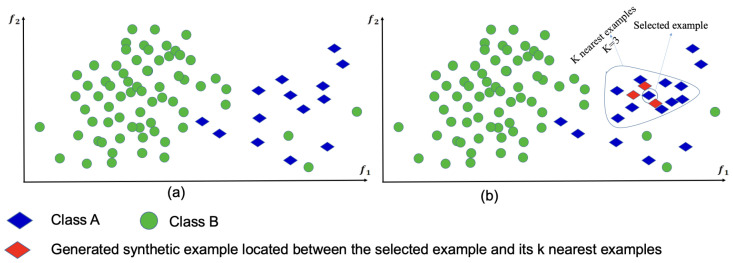
An example of the distribution of a highly imbalanced dataset (**a**) before oversampling and (**b**) after generating synthetic examples using SMOTE.

**Figure 2 ijms-25-02102-f002:**
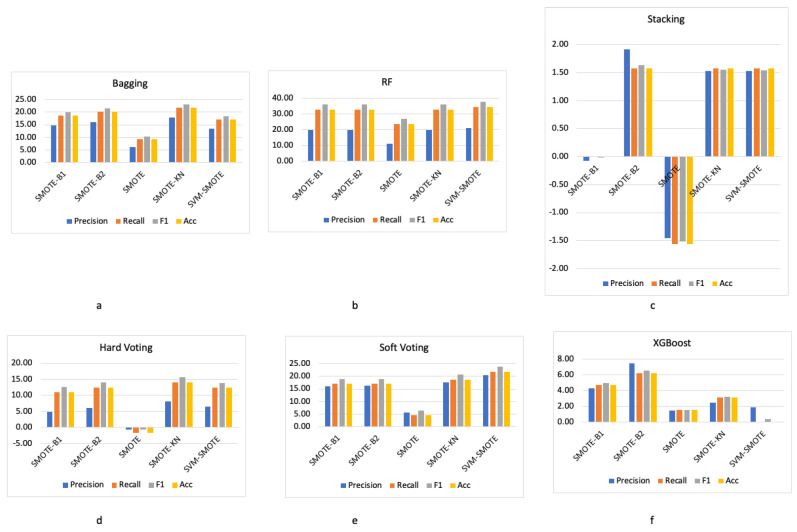
Effects of oversampling techniques along with ensemble learning: (**a**) bagging-based ensemble classifier, (**b**) random forests classifier, (**c**) stacking-based ensemble classifier, (**d**) hard voting-based ensemble classifier, (**e**) soft voting-based ensemble classifier, and (**f**) XGBoost classifier.

**Figure 3 ijms-25-02102-f003:**
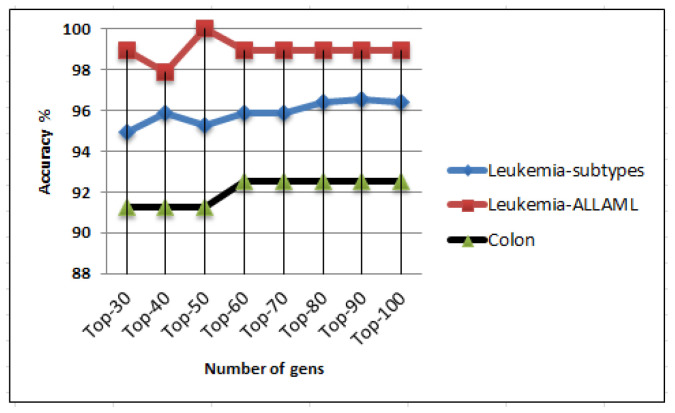
Performance analysis of the random forests classifier with ChiS feature selection on the three datasets with SMOTE preprocessing using 10-fold cross-validation evaluation method.

**Figure 4 ijms-25-02102-f004:**
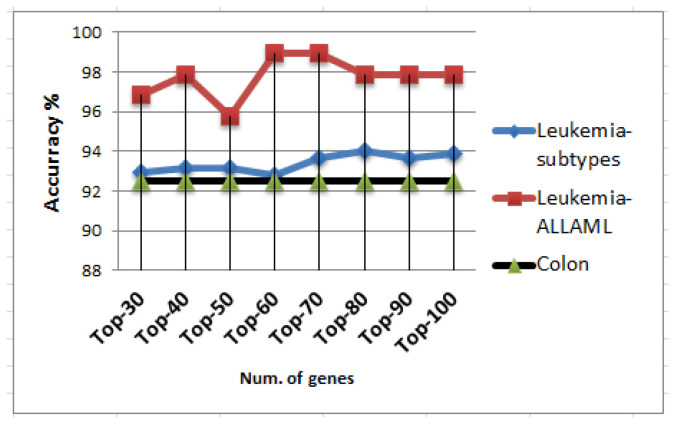
Performance analysis of the ensemble of k-NN classifiers with ChiS feature selection on the three datasets with SMOTE preprocessing using 10-fold crossvalidation evaluation method.

**Figure 5 ijms-25-02102-f005:**
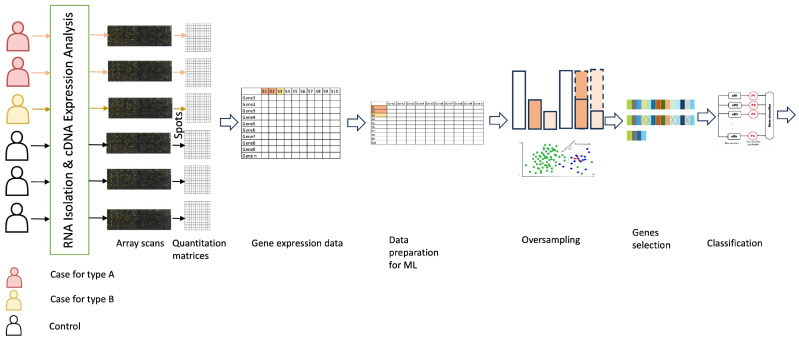
A comprehensive overview for the proposed framework.

**Table 1 ijms-25-02102-t001:** A summary of the related gene expression-based cancer classification approaches.

Ref	Cancer Type	Technique	Feature Selection	Dataset
[5]	Leukemia, colon	FIR-ELM, MLP, ELM, SVM	FGFS	KentRidge
[6]	Lung, colon, breast	ANN: PCA-BEL	PCA	KentRidge
[7]	Colon	SVM	Chi-squar + mRMR	KentRidge, Notterman, E-GEOD-40966
[8]	Colon	SVM, ensemble of SVMs	chi-square, F-Score, PCA, mRMR	KentRidge, BioGPS, Notterman, E-GEOD-40966
[9]	Leukemia, prostate, colon	K-NN, SVM	Fisher, T-Statistics, SNR (Signal-to-Noise Ratio), ReliefF	KentRidge, GEO
[10]	Colon, lymphoma, leukemia	k-NN, BLR, Bayes Net, Neuroblastoma, SVM, MLP, J48, LMT, Random forests.	HDBPSO	
[11]	Colon	Bayes classifier, SVM, BDT		GEO
[12]	Colon	Bayes classifier		Affymetrix Human Genome
[15]	Colon	Bayes classifier	Low entropy filter, and Leave-one-out method	Affymetrix Human Genome
[16]	Breast, Colon, Leukemia, Prostate	GA, ensemble SVM	TCP	KentRidge
[17]	Colon, Lung, Leukemia	SVM, k-NN	MSVDD-RF	KentRidge, Leukemia3, Novartis

**Table 2 ijms-25-02102-t002:** Statistical overview of Leukemia CuMiDa dataset.

Type/Class	Training	Testing	Total	Brief Description
AML	21	5	26	Adult acute myeloid leukemia (AML)
Bone_Marrow	8	2	10	
Bone_Marrow_CD34	6	2	8	They encompass hematopoietic stem and progenitor cells, which possess the capability to develop into all the diverse types of blood cells
PB	8	2	10	
PBSC_CD34	8	2	10	Peripheral blood stem cells (PBSC)
Total	51	13	64	

**Table 3 ijms-25-02102-t003:** Ensemble classifier parameters.

Ensemble Classifier	Paramaters
Bagging Classifier	n_estimators = 10, max_samples = 1.0, max_features = 1.0, bootstrap = True, bootstrap_features = False
Random Forests	n_estimators = 100, criterion = ‘gini’, min_samples_split = 2, bootstrap = True
Stacking	base classifieres: SVM (C = 1.0, kernel = ‘rbf’, degree = 3, gamma = ‘auto’, tol = 0.001, cache_size = 200), Decision Tree (criterion = ‘gini’, splitter = ‘best’, max_depth = None, min_samples_split = 2, min_samples_leaf = 1), Logistic Regression (multi_class = ‘multinomial’, solver = ‘newton-cg’, max_iter = 200). Meta classifier: Logistic Regression
Gradient Boosting	loss = ‘log_loss’, learning_rate = 0.1, n_estimators = 100, subsample = 1.0, criterion = ‘friedman_mse’, min_samples_split = 2, min_samples_leaf = 1
Boosting	booster = gbtree, learning_rate = 0.3, gamma = 0, max_depth = 6, sampling_method = uniform
Hard Voting	base classifiers: SVM (C = 1.0, kernel = ‘rbf’, degree = 3, gamma = ‘auto’, tol = 0.001, cache_size = 200), Decision Tree (criterion = ‘gini’, splitter = ‘best’, max_depth = None, min_samples_split = 2, min_samples_leaf = 1), Logistic Regression (multi_class = ‘multinomial’, solver = ‘newton-cg’, max_iter = 200) voting = ‘hard’
Soft Voting	base classifiers: SVM (C = 1.0, kernel = ‘rbf’, degree = 3, gamma = ‘auto’, tol = 0.001, cache_size = 200), Decision Tree (criterion = ‘gini’, splitter = ‘best’, max_depth = None, min_samples_split = 2, min_samples_leaf = 1), Logistic Regression (multi_class = ‘multinomial’, solver = ‘newton-cg’, max_iter = 200) voting = ‘soft’

**Table 4 ijms-25-02102-t004:** Evaluation results of evaluating oversampling techniques using ensemble learning.

Technique		Evaluation Results			
Ensemble	Oversampling	Precision	Recall	F1	Acc
Bagging	Baseline	79.30	75.00	73.73	75.00
	SMOTE-B1	94.08	93.75	93.64	93.75
	SMOTE-B2	95.43	95.31	95.32	95.31
	SMOTE	85.35	84.38	84.15	84.38
	SMOTE-KN	97.10	96.88	96.71	96.88
	SVM-SMOTE	92.65	92.19	92.13	92.19
RF	Baseline	78.72	65.62	62.35	65.62
	SMOTE-B1	98.58	98.44	98.46	98.44
	SMOTE-B2	98.58	98.44	98.46	98.44
	SMOTE	89.70	89.06	89.14	89.06
	somotKN	98.58	98.44	98.46	98.44
	SVM-SMOTE	**100.00**	**100.00**	**100.00**	**100.00**
Stacking	Baseline	95.49	95.31	95.32	95.31
	SMOTE-B1	95.42	95.31	95.31	95.31
	SMOTE-B2	97.40	96.88	96.95	96.88
	SMOTE	94.03	93.75	93.81	93.75
	SMOTE-KN	97.02	96.88	96.87	96.88
	SVM-SMOTE	97.02	96.88	96.86	96.88
Hard Voting	Baseline	86.46	79.69	78.04	79.69
	SMOTE-B1	91.25	90.62	90.63	90.62
	SMOTE-B2	92.66	92.19	92.19	92.19
	SMOTE	85.78	78.12	77.51	78.12
	SMOTE-KN	94.59	93.75	93.71	93.75
	SVM-SMOTE	92.98	92.19	91.86	92.19
Soft Voting	Baseline	79.43	78.12	76.26	78.12
	SMOTE-B1	95.53	95.31	95.29	95.31
	SMOTE-B2	95.77	95.31	95.29	95.31
	SMOTE	85.08	82.81	82.56	82.81
	SMOTE-KN	97.02	96.88	96.86	96.88
	SVM-SMOTE	**100.00**	**100.00**	**100.00**	**100.00**
XGBoost	Baseline	84.93	84.38	84.08	84.38
	SMOTE-B1	89.20	89.06	89.01	89.06
	SMOTE-B2	92.36	90.62	90.64	90.62
	SMOTE	86.37	85.94	85.65	85.94
	somotKN	87.37	87.50	87.31	87.50
	SVM-SMOTE	86.81	84.38	84.45	84.38

**Table 5 ijms-25-02102-t005:** Statistical overview of Leukemia-subtype.

Type/Class	Training	Testing	Total	Brief Description
BCR-ABL	09	06	15	Leukemias characterized by the presence of B lineage cells that contain t(9;22)
E2A-PBX1	18	09	27	Leukemias characterized by the presence of B lineage cells that contain t(1;19)
Hyperdiploid > 50	42	22	64	A hyperdiploid karyotype
MLL	14	06	20	Rearrangements in the MLL gene on chromosome 11, band q23
T-ALL	28	15	43	T lineage leukemias
TEL-AML1	52	27	79	Leukemias characterized by the presence of B lineage cells that contain t(12;21)
Others	52	27	79	A distinct subgroup of ALL was identified by virtue of its distinctive expression profile
Total	215	112	327	

**Table 6 ijms-25-02102-t006:** Statistical overview of Leukemia-ALLAML and Clon tumor datasets.

Type/Class	Training	Testing	Total
Leukemia-ALLAML dataset			
ALL	27	20	47
AML	11	14	25
Total	38	34	72
Colon tumor dataset			
Tumor	28	12	40
Normal	15	7	22
Total	43	19	62

**Table 7 ijms-25-02102-t007:** The results obtained using MLP for all considered datasets. The highest results shown in **bold**. The table is partitioned into three Sections (1: Individual feature selection techniques before oversampling, 2: Individual feature selection techniques with oversampling, 3: The proposed feature selection technique with and without oversampling).

	Leukemia-Subtype				Leukemia-ALLAML				Colon			
	Prec	Recall	*F*1	Acc	Prec	Recall	*F*1	Acc	Prec	Recall	*F*1	Acc
ChiS	96.80	96.4	96.50	96.43	94.70	94.10	94.00	94.12	84.10	84.20	83.90	84.21
IG	94.80	94.60	94.70	94.64	94.70	94.10	94.00	94.12	84.20	78.90	76.20	78.95
ChiS-SMOTE	96.40	96.40	96.4	96.43	94.70	94.10	94.00	94.12	91.00	89.5	89.00	89.47
IG-SMOTE	95.8	95.50	95.6	95.54	97.20	97.10	97.00	97.06	91.00	89.50	89.00	89.47
ChisIG	**96.80**	**96.40**	**96.50**	**96.43**	**97.20**	**97.10**	**97.00**	**97.06**	87.40	84.20	82.90	84.21
ChisIG-SMOTE	94.80	94.60	94.70	94.64	97.20	97.10	97.00	97.06	**91.00**	**89.50**	**89.00**	**89.47**

**Table 8 ijms-25-02102-t008:** The results obtained using SMO-SVM. The highest results shown in **bold**. The table is partitioned into three Sections (1: Individual feature selection techniques before oversampling, 2: Individual feature selection techniques with oversampling, 3: The proposed feature selection technique with and without oversampling).

	Leukemia-Subtype Dataset				Leukemia-ALLAML Dataset				Colon Dataset			
	Prec	Recall	*F*1	Acc	Prec	Recall	*F*1	Acc	Prec	Recall	*F*1	Acc
ChiS	96.3	95.50	95.7	95.54	88.2	85.30	84.5	85.29	87.4	84.20	82.9	84.21
IG	95.90	95.50	95.60	95.54	**97.20**	**97.10**	**97.00**	**97.06**	81.40	73.7	68.6	73.68
ChiS- SMOTE	95.60	95.50	95.5	95.54	94.70	94.10	94.00	94.12	91.00	89.50	89.00	89.47
IG-SMOTE	96.60	96.40	96.40	96.43	**97.20**	**97.10**	**97.00**	**97.06**	84.10	84.2	83.90	84.21
ChisIG	97.30	96.40	96.60	96.43	94.70	94.10	94.00	94.12	78.9	68.40	59.7	68.42
ChisIG-SMOTE	**97.60**	**97.30**	**97.30**	**97.32**	90.20	88.20	87.8	88.24	**91.00**	**89.50**	**89.00**	**89.47**

**Table 9 ijms-25-02102-t009:** The results obtained using random forests. The highest results shown in **bold**. The table is partitioned into three Sections (1: Individual feature selection techniques before oversampling, 2: Individual feature selection techniques with oversampling, 3: The proposed feature selection technique with and without oversampling).

	Leukemia-Subtype Dataset				Leukemia-ALLAML Dataset				Colon Dataset			
	Prec	Recall	*F*1	Acc	Prec	Recall	*F*1	Acc	Prec	Recall	*F*1	Acc
ChiS	89.00	93.80	91.3	93.75	88.20	85.30	84.50	85.29	79.1	78.9	78	78.95
IG	89.40	93.80	91.3	93.75	90.2	88.2	87.8	88.24	81.4	73.7	68.6	73.68
ChiS-SMOTE	**96.60**	**96.40**	**96.00**	**96.43**	**92.30**	**91.20**	**91.00**	**91.18**	**91.00**	**89.50**	**89**	**89.47**
IG-SMOTE	95.10	94.60	93.9	94.64	88.2	85.30	84.50	85.29	87.40	84.20	82.9	84.21
ChisIG	90	94.60	92.20	94.64	86.40	82.40	81.10	82.35	79.1	78.90	78.00	78.95
ChisIG-SMOTE	94.90	95.50	94.9	95.54	90.20	88.20	87.80	88.24	87.4	84.2	82.9	84.21

**Table 10 ijms-25-02102-t010:** Most prominent gene selection via ChiS feature selection following SMOTE application with random forests.

Dataset	Num. of Genes	Prec	Recall	F1	Acc
Leukemia-subtypes	90	96.5	96.60	96.50	96.56
Leukemia-ALLAML	50	100.00	100.00	100.00	100.00
Colon	60	92.50	92.50	92.50	92.50

**Table 11 ijms-25-02102-t011:** Prominent gene selection via ChiS feature selection following SMOTE application and evaluation using ensemble of k-NN classifiers.

Dataset	Num. of Genes	Prec	Recall	F1	Acc
Leukemia-subtypes	100	94.2	93.90	93.50	93.85
Leukemia-ALLAML	60	99.00	98.90	98.90	98.94
Colon	30	93.10	92.90	92.50	92.95

**Table 12 ijms-25-02102-t012:** Comparisons of our work and related works. SBDNE: similarity-balanced discriminant neighborhood embedding, LLDE: Locally linear discriminant embedding, FIR-ELM: finite impulse response extreme learning machine.

Ref	Techniques	Num. of Genes	Evaluation Mode	Accuracy%
Leukemia-subtype Dataset				
Li Zhang [34]	SBDNE		Holdout method	86.95
Our work	ChisIG-SMOTE and SVM	255	Holdout method	97.32
Our work	ChisIG-SMOTE and SVM	90	10-fold-CV	96.56
Leukemia-ALLAML Dataset				
Li Zhang [34]	LLDE		Holdout method	88.18
Lee et al. [5]	Neural network-based FIR-ELM		10-fold-CV	96.53
Our work	ChiSIG and MLP	145	Holdout method	97.06
Our work	ChiSIG-SMOTE and MLP	145	Holdout method	97.06
Our work	ChiSIG-SMOTE and Random Forests	50	10-fold-CV	100
Colon Tumor Dataset				
Rathore et al. [7]	Sequence of Chi-S and mRMR and SVM	40	10-fold-CV	91.94
Lotfi and Keshavarz [6]	PCA-BEL		10-fold-CV	87.40
Lee et al. [5]	FIR-ELM		10-fold-CV	79.76
Our work	(1) ChisIG-SMOTE and MLP (2) ChisIG-SMOTE and SMO	111	Holdout method	89.47
Our work	ChiS-SMOTE and ensemble of k-NN classifiers	30	10-fold-CV	92.50
Curated Microarray Database (CuMiDa)				
Ilyas et al. [35]	Linear programming, just considering four classes, they considered Bone_Marrow_CD34 and Bone_Marrow as one class		Holdout method	98.44
Our work	PCA, SVM-SMOTE, and RF	50	10-fold-CV	100
Our work	PCA, each of the evaluated oversampling techniques along with RF	50	Holdout method	100

**Table 13 ijms-25-02102-t013:** The majority and minority classes distributions in the training and whole datasets.

Dataset	Training Set		Whole Dataset	
	Majority Class	Minority Class	Majority Class	Minority Class
Leukemia-subtypes	52	9	79	15
Leukemia-ALLAML	27	11	47	25
Leukemia CuMiDa	21	6	26	8
Colon	28	15	40	22

**Table 14 ijms-25-02102-t014:** Count of Genes in the Datasets Pre and Post Feature Selection Techniques.

Technique	Leukemia-Subtype	Leukemia-ALLAML	Colon
Full Genes	12,558	7129	2000
ChiS	300	200	150
IG	300	200	150
ChiSIG	233	145	12
ChiSIG after applying SMOTE	255	163	111

## Data Availability

Data are contained within the article. The datasets used in this study are freely available and can be downloaded by anyone. We have cited the sources of the datasets in the manuscript.

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
