# Peer review of "Gene Expression-Based Cancer Classification for Handling the Class Imbalance Problem and Curse of Dimensionality"

_ijms, 2024, doi:10.3390/ijms25042102_

Round 1

Reviewer 1 Report

Comments and Suggestions for Authors

The authors present a manuscript analysing gene expression-based cancer classification, using SMOTE to address class imbalance problems, and using feature selection to reduce dimensionality. The manuscript is well written, and the issues raised during the previous submission are now dealt with.

Given that the manuscript has already been through two rounds of review, and the authors have addressed the concerns raised, remaining comments are very limited.

Should the authors consider under 'future work' in line 444 whether it would be possible to combine datasets across multiple cancers, or related conditions, in order to better assess specificity (and therefore AUROC and F1 score) when building case-control models? See https://doi.org/10.1016/j.heliyon.2023.e22604 for an interesting recent discussion of the overstatement of specificity in two-class classification problems. Whilst a multi-class ML investigation may be beyond the scope of this work, it would be worth addressing this aspect in future work.

There is a very small number of typos or awkward phrasings, for example 

Line 51, whilst the reference is given as [4] the sentence appears incomplete

Line 407, "it’s evident that" can be removed, the sentence will read better

Line 409, "it’s clear that" can be removed, the sentence will read better

Comments on the Quality of English Language

The English is of good general standard

Author Response

We extend our thanks to the reviewers for their insightful remarks, which have played a crucial role in enhancing the quality of our manuscript.

The authors present a manuscript analysing gene expression-based cancer classification, using SMOTE to address class imbalance problems, and using feature selection to reduce dimensionality. The manuscript is well written, and the issues raised during the previous submission are now dealt with.

Given that the manuscript has already been through two rounds of review, and the authors have addressed the concerns raised, remaining comments are very limited.

Should the authors consider under 'future work' in line 444 whether it would be possible to combine datasets across multiple cancers, or related conditions, in order to better assess specificity (and therefore AUROC and F1 score) when building case-control models? See https://doi.org/10.1016/j.heliyon.2023.e22604 for an interesting recent discussion of the overstatement of specificity in two-class classification problems. Whilst a multi-class ML investigation may be beyond the scope of this work, it would be worth addressing this aspect in future work.

Response: Thanks for reviewer comment and support. In this study we consider both binary-class and multi-class datasets. For binary-class classification dataset, we used Leukemia-ALLAML dataset for ALL and AML classes and also Colon tumor dataset for Tumor and Normal classes. In case of multi-class classification, we also used two datasets namely Leukemia-subtype with 7 classes and Leukemia CuMiDa dataset with 5 classes. Each of the used datasets are evaluated using different evaluation measures including Precision, Recall, F1, and Accuracy.

There is a very small number of typos or awkward phrasings, for example 

  • Line 51, whilst the reference is given as [4] the sentence appears incomplete
    • Response: Sure, thanks for this comment. The current version is updated accordingly.
  • Line 407, "it’s evident that" can be removed, the sentence will read better
    • Response: thanks for this comment, the current version is updated accordingly
  • Line 409, "it’s clear that" can be removed, the sentence will read better
    • Response: thanks for this comment, the current version is updated accordingly

Reviewer 2 Report

Comments and Suggestions for Authors

In the manuscript, the authors address the challenge of classifying cancer based on gene expression, focusing on overcoming class imbalance and the curse of dimensionality. They employ oversampling techniques, notably adding synthetic samples, and propose a novel feature selection method, CHiS&IG, combining Chi-Square and Information Gain techniques. Using four biomedical datasets, the study finds that oversampling and the CHiS&IG method generally improve classification results.

However, the study's reliance on older datasets and microarray technology, which has largely been superseded by RNA-seq, is a significant concern. Additionally, the improvements achieved with oversampling are marginal, especially when compared to existing studies published a decade ago. These factors lead to concerns about the study's current relevance and the practical significance of its findings in the rapidly evolving field of cancer research.

Author Response

we extend our thanks to the reviewers for their insightful remarks, which have played a crucial role in enhancing the quality of our manuscript.

In the manuscript, the authors address the challenge of classifying cancer based on gene expression, focusing on overcoming class imbalance and the curse of dimensionality. They employ oversampling techniques, notably adding synthetic samples, and propose a novel feature selection method, CHiS&IG, combining Chi-Square and Information Gain techniques. Using four biomedical datasets, the study finds that oversampling and the CHiS&IG method generally improve classification results.

However, the study's reliance on older datasets and microarray technology, which has largely been superseded by RNA-seq, is a significant concern. Additionally, the improvements achieved with oversampling are marginal, especially when compared to existing studies published a decade ago. These factors lead to concerns about the study's current relevance and the practical significance of its findings in the rapidly evolving field of cancer research.

Response: Thank you for your insightful comment. We appreciate your suggestion regarding the consideration of data generated using RNA-seq, and we acknowledge this as a valuable direction for our future work. It's important to note that this study primarily focuses on gene-expression datasets utilizing microarray technology. In this version of our manuscript, we specifically incorporate the recently published Leukemia Curated Microarray Database (CuMiDa) from 2019. Furthermore, to ensure the relevance of our findings, we have evaluated our results against a paper published in 2023. Notably, our study reports significantly higher results compared to related studies, as well for the other studies, as detailed in Table 13.

Round 2

Reviewer 2 Report

Comments and Suggestions for Authors

The authors have partially addressed the comments I raised in the last round.